# Anxiety and Depression after Spinal Cord Injury: A Cross-Sectional Study

**DOI:** 10.3390/healthcare12171759

**Published:** 2024-09-03

**Authors:** Brigida Molina-Gallego, María Idoia Ugarte-Gurrutxaga, Laura Molina-Gallego, Fernando Jesús Plaza del Pino, Juan Manuel Carmona-Torres, Esmeralda Santacruz-Salas

**Affiliations:** 1Faculty of Physiotherapy and Nursing of Toledo, Castilla-La Mancha University, Campus Toledo, 45003 Toledo, Spain; brigida.molina@uclm.es (B.M.-G.); juanmanuel.carmona@uclm.es (J.M.C.-T.); esmeralda.santacruz@uclm.es (E.S.-S.); 2Department of Nursing, Primary Health Center (Toledo Area), C/Argentina 19, Madridejos, 45710 Toledo, Spain; lamoga@sescam.jccm.es; 3Faculty of Health Sciences, University of Almeria, 04120 Almeria, Spain; ferplaza@ual.es

**Keywords:** spinal cord injury, psychological disorders, anxiety, depression, acute and chronic

## Abstract

Spinal cord injury (SCI) is a life-changing event that often results in chronic physical damage and challenges in maintaining a good quality of life as it affects every aspect of life. These situations require adjustment, increasing vulnerability to psychological disorders. The objective of this study was to evaluate the impact of SCI on psychological morbidity in individuals with subacute and chronic SCI. The present investigation was designed to determine the presence and extent of psychological complications following SCI. We used two reliable questionnaires and validated psychological assessments to study depression (BDI) and anxiety (STAI), a broad range of factors derived from SCI that may be predictors of certain psychological problems. The psychological assessment revealed alterations in depression and anxiety, although the data do not exceed those of previous investigations. No clear predisposing factors leading to certain psychological pathologies were found. In addition, individuals in the subacute and chronic stages differed in their scores. In individuals with SCI, identifying predictors of psychological problems is difficult, but premature assessment of mental state is essential. This early diagnosis of possible problems or changes at the mental level is fundamental and necessary to avoid possible alterations at the cognitive level and, of course, more serious mental complications.

## 1. Introduction

Spinal cord injury (SCI) occurs when the spinal cord is compressed, scarred, or severed as a consequence of traumatic injury or illness. SCI is associated with the onset of secondary conditions such as chronic pain, inflammation, and chronic weakness, all of which contribute to a poor quality of life and potentially reduced social inclusion [1].

Spinal cord injury (SCI) is a life-changing incident that often results in chronic physical impairment and challenges the maintenance of good quality of life because it affects every aspect of life. Patients are in stressful situations that require adaptation, increasing their susceptibility to psychological disorders [2]. Therefore, individuals with SCI are at greater risk of mental health complications as well. For example, estimates of the prevalence of posttraumatic stress disorder in individuals with SCI range widely from 6% to 44%. For example, depression has been studied extensively among people with spinal cord injury and affects approximately 19–26% of individuals living with SCI; this figure is approximately three times greater than that reported in the general population [3,4]. Depressive symptoms are associated with innumerable negative consequences among people with SCI, including lower functional independence, more secondary complications, poorer community and social integration, and lower self-appraised health [5].

Anxiety is problematic for adults with acquired spinal cord injury (SCI), with up to 45% of injured individuals reporting extreme discomfort, panic, or terror, contributing to a high risk of experiencing disorders such as generalized anxiety (GAD) [6]. On many occasions, this is caused by the traumatic nature of the SCI. Anxiety problems can also originate from the fear of the complications that the spinal cord injury involves (for example, autonomic dysreflexia, ulcers) and, of course, the psychological state prior to the spinal cord injury [7,8].

Discrepancies in estimates of depression and anxiety after SCI may, in part, be explained by the different definitions and subsequent measurements used, which have been criticized for different conceptual and psychometric reasons. Given the fundamental role of depression and anxiety in the health and well-being of people with SCI, early and accurate detection and measurement of these two parameters are essential for advancing both research and clinical practice [9,10].

Rates of anxiety, substance abuse, and other mental health problems in individuals with SCI also tend to be higher than those reported in the general population [11].

It can normally be assumed that a greater degree of disability or severity of SCI is correlated with higher levels of depression and anxiety and therefore leads to greater difficulty in coping with these individuals. However, through clinical observations and previous research, evidence suggests that this is not the case [12].

On the other hand, different investigations suggest that SCI patients have higher rates of depression and anxiety than the general population does, and these rates are comparable to those reported several years after injury [13].

To our knowledge, no previous studies have been carried out on depression and anxiety in SCI patients in Spain. Therefore, we believe that there is a need to investigate this topic further.

The aim of this study was to evaluate the prevalence of depression and anxiety after SCI and its associations with sociodemographic and health variables.

## 2. Materials and Methods

This was a cross-sectional study in which anxiety and depression scores in both groups (subacute and chronic stages) were matched with different variables, such as sex, severity level, and extent of injury. The participants with SCI were recruited from the “National Hospital for Paraplegics”, an SCI rehabilitation hospital situated in the city of Toledo, Spain.

### 2.1. Participants

Among the participants with SCI, 50 were included in the subacute stage group, with a mean age of 46.82 (SD 15.77) for 20 women and 30 men, and 50 were included in the chronic stage group, with a mean age of 47.80 (SD13.75) for 19 women and 31 men. This cross-sectional study was part of a larger project approved by the Ethics Committee of the Toledo Area (N° CEIS-73, 5 June 2013). The committee belongs to the Toledo Health Area, not the same hospital.

In the subacute phase, we included participants with a recent first admission to our spinal cord injury unit, with a time from injury fluctuating between 4 and 6 months. Participants with a time from injury of at least 1 year who attended the hospital for annual follow-up were included in the chronic phase.

Inclusion criteria for both groups included (1) presence of SCI; (2) injury level less than or equal to C4; (3) age at injury 18 years or older; (4) age at interview 18–85 years; and (5) Spanish-speaking.

Exclusion criteria were as described below: (1) no evidence of SCI; (2) a level greater than C4; (3) an age of injury lower than 18 years; (4) the presence of clinically evident traumatic brain injury (TBI) (due to the known association of TBI with cognitive dysfunction); (5) severe psychiatric disorders; (6) a history of central or peripheral neurological problems prior to SCI; and (7) a known history of alcohol or drug abuse.

### 2.2. Clinical and Demographic Data

Demographic information was obtained from each participant, including age, sex, education (literacy and primary, secondary, and university education). Additionally, each participant was interviewed about a history of frequent alcohol and/or drug use [14,15].

Clinical information included lesion level (cervical, dorsal, and lumbar), American Spinal Injury Association Impairment Scale (AIS) grade (A, B, C, D, E) and time since injury (months).

### 2.3. Emotional Status Assessment

As we have already noted, depression and anxiety are two factors to consider after SCI. Therefore, our study focuses on these two variables and how they are influenced or can be influenced by other prominent factors within the SCI.

In this study, anxiety and depression were considered important because of their significance; both were explored through tests that assessed emotional state.

For this reason, we evaluated depression and anxiety via the Beck Depression Inventory (BDI) and State Anxiety Inventory (STAI) [16,17,18].

#### 2.3.1. Beck Depression Inventory (BDI)

While there is some indication that the BDI may inflate estimates in individuals with SCI because of some somatic-based items, it has been shown to be generally reliable with samples of people with SCI [19].

The BDI is a widely used measure to evaluate the degree of severity of depressive symptoms. This questionnaire contains 21 items that assess both the somatic and affective aspects of depression, with four answer choices per item. For each item, the patient must select the statement that best fits their feelings during the last two weeks. They are defined according to the severity and intensity of the symptom and are listed in order from greatest to least serious. Each item is given 0 to 3 points depending on the alternative chosen, and the total score of the test can range from 0 to 63 points.

The higher the total score is, the greater the degree of depressive symptoms. Cutoff points were established to differentiate between severe depression and moderate or mild depression, and scores > 17 were indicative of moderate depression.

The BDI seems to have acceptable psychometric properties as an instrument for evaluating depressive symptoms in adults in the general Spanish population.

#### 2.3.2. State Anxiety Inventory (STAI)

The STAI questionnaire comprises separate self-assessment scales that measure two aspects related to anxiety. Anxiety is a specific state (S), and anxiety is a more stable trait (T) [20,21].

There are 2 subscales within this measure. First, the State Anxiety Scale (S-Anxiety) evaluates the current state of anxiety, asking how respondents feel “right now”, using items that measure subjective feelings of apprehension, tension, nervousness, worry, and activation/arousal of the autonomic nervous system. The Trait Anxiety Scale (T-Anxiety) evaluates relatively stable aspects of “anxiety proneness”, including general states of calmness, confidence, and security.

The STAI has 40 items, with 20 items allocated to each of the S-Anxiety and T-Anxiety subscales. A cutoff value of 39–40 was suggested for the ability of the STAI to detect clinically significant symptoms on the scale. The range of scores for each subtest is 20–80, with higher scores indicating greater anxiety.

### 2.4. Statistical Analysis

The data were analyzed via SPSS, version 29.0 (IBM Corp., Armonk, NY, USA; license from the University of Castilla—La Mancha).

The data analysis was carried out in several phases:

First, a descriptive analysis was performed. Qualitative variables are expressed as counts (N) and percentages (%), and quantitative variables are expressed as arithmetic means (Ms) and standard deviations (SDs).

An inferential analysis was subsequently performed. For categorical variables, comparisons were made between groups via Pearson χ^2^, Fisher’s exact, or likelihood ratio tests for categorical variables (sex, educational level, lesion level, and AIS). Student’s *t* test was used for quantitative variables (age, time since injury, BDI score, STAI-T score, and STAI-S score).

Finally, a Pearson correlation was performed for the scores of the two scales used. In addition, to control for the influence of sex and age, a partial correlation was performed.

All hypothesis tests were two-sided. Values with a confidence level of 95% (*p* < 0.05) were considered statistically significant.

### 2.5. Ethical Considerations

The study was carried out after authorization was obtained from the Ethics Committee of the Toledo Area. In addition, informed consent was obtained from the participants in this study. In terms of confidentiality, each patient was assigned a code.

## 3. Results

The characteristics of the 100 participants with SCI (50 subacute and 50 chronic) are summarized in Table 1. It can be seen that there were no significant differences between the participants with regard to age, gender, educational level, or lesion level. In both samples, AIS differed significantly (χ^2^ = 8.23, *p* = 0.041), with a higher proportion of AIS-A (n = 25) and AIS-D (n = 15). Time since injury was significantly longer in chronic than in subacute SCI (unpaired *t* = −8.24 *p* < 0.001).

In the subacute SCI group, 90% of participants took at least one neuroactive medication, whereas in the chronic SCI group this percentage was 74% (χ^2^ = 4.33 *p* = 0.037). In relation to smoking habits, significant differences were observed between the two groups. (χ^2^ = 20.94 *p* = 0.001).

In our study, the higher the total BDI score was, the greater the degree of depressive symptoms. Cutoff points were established to differentiate between severe depression and moderate or mild depression, and scores > 17 were indicative of moderate depression. In the subacute group, 4% of the patients scored above this value, and in the chronic group, 16% scored above this value.

Concerning anxiety, in our study, we used the STAI, and a cutoff value of 39–40 was suggested to detect clinically significant symptoms. In the subacute group, 8% of the STAI-T participants and 10% of the STAI-S participants scored over 40, and in the chronic group, both the STAI-T and the STAI-S scores were 14%.

Considering emotional status (Table 2), differences were observed between groups. In the table, we cross the values of anxiety and depression in both groups, relating them to the cause of the same (traumatic or medical).

The most important difference was the tendency toward a more depressive mood (BDI) in patients with traumatic SCI in both the subacute and chronic groups. However, when the cause of the injury is medical, the scores are lower. The differences were significant in the chronic group (*p* = 0.021) when compared by cause of injury (traumatic or medical) but not when we compared the subacute group.

In terms of anxiety, the scores were higher in patients whose cause of injury was traumatic than in those whose cause of injury was medical. The differences did not appear within the groups when we compared the cause of injury. In terms of trait anxiety (STAI-T), no statistically significant differences were detected.

When we cross the scores of the emotional state variables with sex (Table 3), significant differences are observed in the chronic group in the STAI-S (*p* = 0.028) and also in the subacute and chronic groups (*p* = 0.041). For the rest of the scores, the differences were not statistically significant, although the group of women obtained higher scores in both the subacute and chronic groups on the STAI-T and BDI.

Table 4 and Table 5 show the scores on the scales according to the extent and severity of the SCI. Table 4 shows the differences between complete and incomplete lesions at the motor level. No significant differences were observed between the groups. In the BDI, the scores were higher for incomplete lesions in the acute phase than for incomplete lesions, which changed the scenario in the chronic stage of the injury. With respect to anxiety, the scores do not follow a fixed rule, although the differences are not statistically significant.

In Table 5, we observe differences in the groups according to the extent of the injury; in the BDI, scores are higher in paraplegics in the acute stage, changing to tetraplegia in the chronic stage. In terms of anxiety, the highest scores were obtained by the paraplegic group in the chronic stage, although the difference was not significant.

Subsequently, correlation analyses were carried out between the scales (STAI-S, STAI-T, and BDI).

In Table 6, the results revealed a significant correlation between the scales STAI and BDI. Finally, in Table 7, the results of partial correlation, where the influence of the variables sex and age was controlled, show moderate correlations, but all of them are statistically significant.

To conclude, logistic regression was performed on the findings. The variables associated with suffering from state anxiety (STAI-S) in people with spinal cord injuries were traumatic (OR = 15.591 *p* = 0.036), lumbar (OR = 30.067, *p* = 0.009), and the consumption of psychotropic drugs (OR = 19.803, *p* = 0.048). Table 8 shows these results.

## 4. Discussion

Spinal cord injury is associated with acute complications affecting the motor, sensory, cardiovascular, thermoregulatory, bronchopulmonary, urinary, gastrointestinal, and genitourinary systems, which may be life-threatening and/or prolong rehabilitation. It is essential to understand these complications in the acute phase to manage them appropriately. These complications, both subacute and chronic, are common and have a negative impact on patients’ functional independence and quality of life, causing significant functional, psychological, and socioeconomic disability, all resulting in significant psychological disorders in these patients [22,23,24].

In general, 40% of patients received at least one mental disorder diagnosis from an SCI team psychologist during outpatient evaluations. Consistent with previous studies, mental disorders were associated with lower life satisfaction and greater impairment in daily activities from health and mental health problems. These findings highlight the prevalence of mental disorders and the associated reduction in quality of life for many patients with SCI [25,26].

Importantly, among the psychiatric disorders that occur after SCI, the most frequently diagnosed disorder is depression. In our study, the higher the total BDI score was, the greater the degree of depressive symptoms. Cutoff points were established to differentiate between severe depression and moderate or mild depression, and scores > 17 were indicative of moderate depression. In the subacute group, 4% of the patients scored above this value, and in the chronic group, 16% scored above this value. In several studies, higher scores were observed on this scale, although there were contradictory results since some studies have shown that spinal cord injury does not have to lead to a depressive state. For example, in a meta-analysis, the prevalence of depression among persons with SCI was estimated to be 22.2% [4].

An earlier study in Sri Lanka revealed that 41% of persons with SCI had depression according to a screening tool [27]. In a study in which the Beck Depression Questionnaire (BDI) was used to measure depressive symptoms during inpatient rehabilitation, approximately 60% of people with spinal cord injury never scored in the depression range, and 40% reported having depressive symptoms that were important on at least one occasion [28].

Subsequent studies, such as that by Pollard and Kennedy published in 2007 on 87 SCI patients with traumatic etiology, revealed that two-thirds of the sample did not present signs or symptoms of depression. It was concluded that most people affected by SCI have sequelae or consequences of their disability without significant levels of psychopathology [29].

For anxiety, in our study, we used the STAI, and a cutoff value of 39–40 was suggested to detect clinically significant symptoms on the scale. Different studies have estimated that the prevalence of anxiety on the basis of self-reports is 15–32% [2,17]. Our study revealed that anxiety in the subacute group was 8% on the STAI-T score and 10% on the STAI-S score, and in the chronic group, both the STAI-T score and the STAI-S score were 14%.

In most studies that investigate anxiety, the phenomena of anxiety and/or depression are implicit. Clinically, determining the difference is very difficult since they are interrelated processes [30], although the trend in recent publications is to consider that one does not always have to address a situation of postinjury anxiety and depression [31,32].

The objective of this study was to evaluate the level of depression and anxiety related to different sociodemographic and clinical variables, such as sex, age, level, severity of injury, and time since injury, and to predict which factors may influence depression and/or anxiety after SCI.

First, we compared groups according to the extent of the injury (paraplegia versus tetraplegia) and complexity (complete motor injury and incomplete).

It would seem obvious that people with tetraplegia and those with complete motor injuries would be more depressed and anxious due to their greater mobility restrictions, but this was not observed in this study. No differences were observed in this study, as shown in the tables.

The results revealed that there were no statistically significant differences between any of the groups. In addition, people with paraplegia or incomplete motor injuries had higher average scores for depression and anxiety than people with tetraplegia and complete injuries. Previous research also supports these findings [12,33,34].

Next, we wanted to determine the relationships that could exist between the cause of the injury and the different psychological variables (anxiety and depression). Then, we compared differences between groups according to cause of injury (traumatic versus medical). There were no significant differences in these psychological factors (depression and anxiety), although traumatic injury scores were higher than medical injury scores were. Traumatic injuries most often involve a traumatic experience, such as a traffic accident or fall.

These results coincide with those of previous studies, which highlight that anxiety and depression problems are often due to the nature of the cause of the injury [2,6].

When we observe these differences in psychological variables from a gender perspective, the results of this study do not confirm significant differences in these scales depending on the sex of the individuals with SCI. Women score higher on the BDI and STAI in the subacute and chronic groups, with only statistically significant differences in state anxiety (STAI-S) in the chronic group; the remaining groups do not significantly differ [35].

Relatedly, some studies address the components of extraversion, neuroticism, depression, and anxiety in men and women with spinal cord injury and conclude that women are more prone to distress emotional states, depression, and anxiety. However, numerous studies do not find differences between men and women in the psychological scales analyzed.

The need to review and guide the study of this variable has increased given its importance in the rehabilitation process of people with spinal cord injury since this variable has frequently been ignored [35,36,37,38].

Previous studies have investigated various demographic and spinal cord injury factors that may have an influence on psychological variables after spinal cord injury. However, there is no consensus on the association of demographic factors with psychological variables. Studies have shown that depression is associated with female sex, tetraplegia, low educational level, and having a family member as a caregiver [39].

However, in another study of 849 patients with depression and SCI, Bombardier et al. reported that demographic factors, such as gender and education level, and injury-related factors, such as level and severity of injury, were not significantly associated with depression or anxiety [14,25]. Research by Lee et al. also revealed that marital status and age were associated with suicide risk, whereas other demographic factors were not significantly correlated with psychological variables. In relation to spinal cord injury, lower levels of injury are associated with greater voluntary movement and expectations of independence, but the level and extent of injury are not related to psychological variables [2].

A correct mental state is necessary for the success of rehabilitation, as well as for the ability to perform activities of daily living according to one’s disability. It is also essential for the planning and solving of problems to achieve the right social adaptation and, consequently, progress in their care, which will help improve their quality of life.

## 5. Conclusions

### 5.1. Conclusions from Our Research in Relation to Depression

In the subacute group, the paraplegics score was greater than that in the tetraplegic group, and these results were reversed in the chronic group.

Whereas we compared complete and incomplete motor injuries, incomplete motor injuries were associated with higher depression scores in patients in the subacute group.

In terms of the cause of injury, the results are more conclusive, with the traumatic group scoring higher in terms of depression in both groups (subacute and chronic).

Finally, the female group scored higher in terms of depression in both the acute and chronic groups.

We conclude that for depression, the differences between the subacute and chronic groups were significant when compared by cause of injury, but these differences were not significant when compared by sex, extent of injury, or level of injury.

### 5.2. Conclusions Regarding Anxiety

The paraplegia group had higher STAI-S scores in the subacute and chronic groups. The STAI-T does not correlate at all.

Compared with the extent of the injury (complete or incomplete), the scores are higher for state anxiety (STAI-S) in the group with complete injuries, both in the acute and chronic groups. On the other hand, in trait anxiety (STAI-T), these results are reversed, and the group with the highest scores is the incomplete injury group.

In relation to the cause of the injury, both types of anxiety (trait and state) had higher scores in the traumatic cause group and in the subacute and chronic groups.

Compared with men, women with state anxiety scored higher in both the acute and chronic groups.

It is interesting to compare the results between the two questionnaires used in our research. This study revealed that depression is related to anxiety in its two measured variants (state anxiety and trait anxiety). In fact, there is a strong positive correlation between the BDI and the STAI. This is not a cause-effect explanation but a correlational expression in which the results of these two questionnaires move in parallel in the same direction, more plus and less minus. As seen from both the population data and the correlations, the association between state anxiety and depression is strongest, as can be deduced from the measured scores. Research over the years has used these scales to measure anxiety and depression in this population, which, as we have previously reported, has specific characteristics [19,40,41].

The results of this quantitative review confirm that a significant minority of people with SCI experience disorders or symptoms of depression and anxiety that persist over time. The findings highlight the importance of routine psychological assessment throughout the spinal rehabilitation process, as well as the need for consistency in the selection and administration of psychological instruments across studies. Further research into the utility and validity of individual measures of psychological function following SCI, including the appropriateness of different cut-off scores, will help to establish guidelines for the interpretation of depression and anxiety outcomes in this specific population. Addressing these issues will improve the clinical utility of this research.

In conclusion, we believe that early assessment of the mental or psychological state of SCI patients is essential. As we have shown in our study, identifying predictors of psychological problems in SCI patients is difficult. This early diagnosis of possible problems or changes at the mental level is fundamental and necessary to avoid possible alterations at the cognitive level and, of course, more serious mental problems.

## Figures and Tables

**Table 1 healthcare-12-01759-t001:** Demographic and clinical characteristics of patients with subacute and chronic spinal cord injury.

		Subacute n = 50	Chronic n = 50	*p*-Value
Age Mean (M; SD)		46.82 (15.77)	47.80 (13.76)	0.493
Sex n (%)	Men Women	30 (%) 20 (%)	31 (%) 19 (%)	0.838
Education n (%)	Read/Write Primary Secondary University	3 (%) 21 (%) 19 (%) 7 (%)	5 (%) 25 (50%) 12 (%) 8 (%)	0.066
Cause of Injury	Traumatic Medical	32 (%) 18 (%)	34 (%) 16 (%)	0.673
Injury Level n (%)	Cervical Dorsal Lumbar	22 (%) 23 (%) 5 (%)	15 (%) 28 (%) 7 (%)	0.406
AIS n (%)	A B C D	12 (%) 7 (%) 16 (%) 15 (%)	25 (%) 7 (%) 8 (%) 10 (%)	0.041 *
Smoking habits n (%)	Smoker Nonsmoker Ex-smoker	13 (%) 21 (%) 16 (%)	12 (%) 38 (%) 0	<0.001 *
Alcohol Consumption n (%)	Drinker Non drinker	14 (%) 36 (%)	10 (%) 40 (%)	0.35
Consumption of Drugs n (%)	Yes Non	4 (%) 46 (%)	3 (%) 47 (%)	0.695
Consumption of Psycho-pharmaco n (%)	Yes Non	45 (%) 5 (%)	37 (%) 13 (%)	0.037 *
Time since Injury (Months) Mean (M; SD)		5.74 (2)	124.42 (101.78)	<0.001 *

* *p* < 0.005.

**Table 2 healthcare-12-01759-t002:** Emotional status of patients with subacute and chronic spinal cord injury with the cause of injury.

	Subacute			Chronic		
	Traumatic n = 32	Medical n = 18	*p*-Value	Traumatic n = 34	Medical n = 16	*p*-Value
BDI (M; SD)	8.56 (7.70)	6.50 (4.43)	0.186	9.76 (8.25)	5.88 (4.5)	0.021 *
STAI-T (M; SD)	22.38 (13.25)	18.67 (9.4)	0.132	19.26 (13.54)	17.50 (12.95)	0.353
STAI-S (M; SD)	20.56 (11.40)	19.11 (7.76)	0.129	22.88 (13.47)	18.25 (10.94)	0.102

* *p* < 0.005.

**Table 3 healthcare-12-01759-t003:** Emotional status of patients with subacute and chronic spinal cord injury stratified by sex (women/men).

	Subacute			Chronic		
	Women n = 20	Men n = 30	*p*-Value	Women n = 19	Men n = 31	*p*-Value
BDI (M; SD)	8.00 (4.70)	7.70 (7.87)	0.130	10.32 (8.64)	7.42 (6.52)	0.053
STAI-T (M; SD)	19.30 (10.65)	22.20 (12.94)	0.102	24.26 (15.17)	15.29 (10.82)	0.080
STAI-S (M; SD)	21.45 (10.19)	19.10 (10.23)	0.877	24.84 (13.72)	19.29 (11.92)	0.028 *

* *p* < 0.005.

**Table 4 healthcare-12-01759-t004:** Emotional status of spinal cord injury in subacute and chronic patients with complete motor and incomplete motor injuries.

	Subacute			Chronic		
	Complete Injuries n = 12	Incomplete Injuries n = 38	*p*-Value	Complete Injuries n = 25	Incomplete Injuries n = 25	*p*-Value
BDI (M; SD)	6.08 (4.99)	8.37 (7.16)	0.225	8.56 (6.79)	8.48 (8.19)	0.647
STAI-T (M; SD)	19.83 (9.11)	21.42 (12.91)	0.156	18.56 (13.83)	18.84 (12.93)	0.343
STAI-S (M; SD)	20.67 (11.09)	19.84 (10.03)	0.676	21.92 (11.87)	20.88 (13.88)	0.839

**Table 5 healthcare-12-01759-t005:** Emotional status of spinal cord injury in subacute and chronic patients with paraplegia and tetraplegia.

	Subacute			Chronic		
	Tetraplegia n = 22	Paraplejia n = 28	*p*-Value	Tetraplegia n = 15	Paraplegia n = 35	*p*-Value
BDI (M; SD)	7.05 (4.27)	8.43 (8.19)	0.144	8.87 (9.14)	8.37 (6.74)	0.260
STAI-T (M; SD)	21.45 (12.74)	20.71 (11.69)	0.643	15.27 (11.51)	20.17 (13.82)	0.133
STAI-S (M; SD)	19.73 (8.43)	20.29 (11.52)	0.203	17.87 (11.06)	22.91 (13.32)	0.152

**Table 6 healthcare-12-01759-t006:** Correlations between the different subscales of anxiety and depression.

	STAI-T	STAI-S	BDI
STAI-T	-	0.733 **	0.598 **
STAI-S		-	0.768 **
BDI			-

** Correlation is significant at the 0.01 bilateral level.

**Table 7 healthcare-12-01759-t007:** Partial correlations controlling for sex and age in different subscales of anxiety and depression.

	STAI-T	STAI-S	BDI
STAI-T	-	0.729 **	0.593 **
STAI-S		-	0.768 **
BDI			-

** Correlation is significant at the 0.01 bilateral level.

**Table 8 healthcare-12-01759-t008:** Factors associated with the STAI-S.

Variable	OR (IC 95%)	*p*
Cause of Injury		
Traumatic	15.591 (1.198–202.937)	0.036 *
Non-Traumatic	Reference	
Level of Injury		
Cervical	Reference	
Dorsal	1.526 (0.325–7.160)	0.592
Lumbar	30.067 (2.371–381.272)	0.009 *
Consumption of Psychotropic Drugs		
No	Reference	
Yes	19.803 (1.029–381.237)	0.048 *

* *p* < 0.005.

## Data Availability

The data are held in confidence by the lead author of this manuscript. They can be made available on request.

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
