# Peer review of "Anxiety and Depression after Spinal Cord Injury: A Cross-Sectional Study"

_healthcare, 2024, doi:10.3390/healthcare12171759_

Round 1

Reviewer 1 Report

Comments and Suggestions for Authors

The study focuses on the psychological impact of spinal cord injury (SCI) on individuals, particularly in terms of depression and anxiety, at subacute and chronic stage. For that, the study used two reliable and validated psychological assessment tools to identify factors derived from SCI that might predict psychological problems.: BDI (Beck Depression Inventory) and STAI (State-Trait Anxiety Inventory).

The paper suggests that while it is challenging to identify predictors of psychological problems in individuals with SCI, early mental health assessment is crucial. Early diagnosis can help in addressing potential psychological issues before they develop into more serious mental health conditions.

Nonetheless, minor changes need to be addressed before the study is accepted.

1.     The entire study needs to improve the English writing. Sometimes is difficult to follow the text. Authors could use a research editor to help with the text.

2.     Please rephrase the first two sentences of abstract for clarity and flow: (line 15)-17.

3.     Also, this reviewer recommends focusing the both the title and the abstract on anxiety and depression, and not mental health problems (which is a bit generalist as all methodology was directed to studying depression and anxiety).

4.     In Methods (page 2), Include the city and country of the hospital. Also include the number of the Ethical Committee and describe if it was from the same hospital (weird the statement of “Toledo area”). In section 2.1., first paragraph, there is a number in parenthesis next to the mean age of the participants. What does it mean?? Is it the s.d. or s.e.m.??? Please describe.

5.     Page 313, please, use more specific diagnostic terms for “women are more neurotic”, the phrase as it is does not reads nicely…. Authors could change to, for example, “women are more prone to distress emotional states…” or another more specific terminology.  In the modern psychology, the terminology neurotic is less commonly used as a formal diagnosis.  

6.     In discussion section, authors should include a paragraph about physiological changes after SCI that could be leading to the mental health problems. Also, as chronic pain is observed in many SCI patients, is there a correlation between the patients and their level of pain?? 

Comments on the Quality of English Language

The entire study needs to improve the English writing. Sometimes is difficult to follow the text. Authors could use a research editor to help with the text.

Author Response

Comments 1: The entire study needs to improve the English writing. Sometimes is difficult to follow the text. Authors could use a research editor to help with the text.

Response 1: The manuscript has been reviewed by a native colleague.

Comments 2: Please rephrase the first two sentences of abstract for clarity and flow: (line 15)-17.

Response 2: We agree with this comment, therefore, we have changed this phrase.

Comments 3: Also, this reviewer recommends focusing the both the title and the abstract on anxiety and depression, and not mental health problems (which is a bit generalist as all methodology was directed to studying depression and anxiety).

Response 3: We agree with this comment, therefore, we have changed the title.

Comments 4:  In Methods (page 2), Include the city and country of the hospital. Also include the number of the Ethical Committee and describe if it was from the same hospital (weird the statement of “Toledo area”). In section 2.1., first paragraph, there is a number in parenthesis next to the mean age of the participants. What does it mean?? Is it the s.d. or s.e.m.??? Please describe.

Response 4: We agree with this comment, therefore, we have added city and country of the hospital also we have included the number of the ethical committee. And we have explained the meaning of the parenthesis, it is the standard deviation.

Comments 5: Page 313, please, use more specific diagnostic terms for “women are more neurotic”, the phrase as it is does not reads nicely…. Authors could change to, for example, “women are more prone to distress emotional states…” or another more specific terminology.  In the modern psychology, the terminology neurotic is less commonly used as a formal diagnosis.  

Response 5: We agree with this comment, we have changed the term.

Comments 6: In discussion section, authors should include a paragraph about physiological changes after SCI that could be leading to the mental health problems. Also, as chronic pain is observed in many SCI patients, is there a correlation between the patients and their level of pain?? 

Response 6: We agree with this comment, we have included a paragraph about physiological changes after SCI.

Reviewer 2 Report

Comments and Suggestions for Authors

I congratulate you for your interesting research on this topic, really important.

Some recommendations, questions and comments:

1)      In Inclusion Criteria it appears "injury level below C4"; in Exclusion Criteria "injury level above C4". How to classify CASES WITH INJURY LEVEL C4? I consider it necessary to put "injury level greater than or equal...", or "less than or equal...", for example.

2)       Where it says, on line 97: (4) the presence of clinically demonstrated TBI. Have they previously put the full expression, with its abbreviation TBI in parentheses?

3)      IMPORTANT: In lines 182-184, it seems that they are comments from the draft of the manuscript, which should be DELETED: "This section may be divided by subheadings. It should provide a concise and precise description of the experimental results, their interpretation, as well as the experimental conclusions that can be drawn".

4)      In lines 194-6 it says: "The differences were significant in the chronic group (p = 0.021) when compared by cause of injury (traumatic or medical) but not when we compared the subacute and chronic group." But I think the statistical analysis comparing the subacute and chronic group does not appear. Therefore, perhaps the correct text as a comment on the table would be: ""The differences were significant in the chronic group (p = 0.021) when compared by cause of injury (traumatic or medical) but not when we compared the subacute group".

5)      On lines 198-9, it says: "Statistically significant were the results when we compared groups (subacute and chronic) in State Anxiety (STAI-S)." WHAT STATISTICAL ANALYSIS DO YOU BASED ON TO STATE THIS? (I THINK IT DOES NOT APPEAR IN TABLE 2).

6)      In lines 224-5 it says: "About anxiety, it seems that the highest scores are obtained by the paraplegic group in both the acute and chronic stages." But the data do not support this statement in the acute phase, so the correct phrase should refer only to the chronic phase: "About anxiety, it seems that the highest scores are obtained by the paraplegic group in the chronic stage, although not significantly."

7)      On line 233, it would be advisable to change the wording of the phrase: "Tables 6, 7 The results revealed a..." For example, there is a missing period followed by 7.

8)      In Table 8, values ​​of p<0.05 could have an asterisk*, as in Table 1.

9)      IMPORTANT: On lines 239-240, it says: "The variables that are associated with suffering from state anxiety (STAI-S) in spinal cord injured people were that the cause of the injury was non-traumatic (OR=15.591... "   But it really seems that it correlates with TRAUMATICS. Therefore, I think the text should be: “The variables that are associated with suffering from state anxiety (STAI-S) in spinal cord injured people were that the cause of the injury was traumatic (OR=15.591..."

10)   In discussion, data appears that have not been previously clearly stated in Results. For example, on lines 272-3: "Our study shows results like those found, anxiety in the subacute group is at 8% STAI-T and 10% in STAI-S, and in the chronic group both STAI-T and STAI-S are at 14%".   

I propose that said data should previously appear in the Results section, and then be commented on in Discussion.

11)   IMPORTANT: Lines 279-282 should be DELETED (they are comments from the authors' draft): "Authors should discuss the results and how they can be interpreted from the perspective of previous studies and of the working hypotheses. The findings and their implications should be discussed in the broadest context possible. Future research directions may also be highlighted".

12)       IMPORTANT: In lines 334-5, in Conclusions, where it says: "The group of paraplegics and tetraplegics score higher than the group of subacute paraplegics, and these results are reversed in the chronic group".

It should say: "In the subacute group, the group of paraplegics score higher than the group of tetraplegics, and these results are reversed in the chronic group".

13) In lines 337-8, it says: "and this relationship is reversed in the chronic group". I think that final sentence should be removed, because the scores between both groups in the chronic cases are almost equal."

14) In conclusions 1, on Depression, I think it should specify which differences are significant and which are not.

15) The last paragraph of the Conclusions is not derived from the results of this research and, therefore, I think it should appear in Discussion: "A correct mental state is necessary for the success of rehabilitation as well as for the ability to perform activities of daily living according to their disability. It is also essential for the planning and solving of problems in order to achieve a right social adaptation and, consequently, progress in their care, which will help to improve their quality of life."

Author Response

Comments 1: In Inclusion Criteria it appears "injury level below C4"; in Exclusion Criteria "injury level above C4". How to classify CASES WITH INJURY LEVEL C4? I consider it necessary to put "injury level greater than or equal...", or "less than or equal...", for example.

Response 1: We agree with this comment, therefore, we have changed this phrase.

Comments 2: Where it says, on line 97: (4) the presence of clinically demonstrated TBI. Have they previously put the full expression, with its abbreviation TBI in parentheses?

Response 2: We agree with this comment, therefore, we have put the full expression.

Comments 3:  IMPORTANT: In lines 182-184, it seems that they are comments from the draft of the manuscript, which should be DELETED: "This section may be divided by subheadings. It should provide a concise and precise description of the experimental results, their interpretation, as well as the experimental conclusions that can be drawn".

Response 3: We agree with this comment, therefore, we have deleted this paragraph.

Comments 4:  In lines 194-6 it says: "The differences were significant in the chronic group (p = 0.021) when compared by cause of injury (traumatic or medical) but not when we compared the subacute and chronic group." But I think the statistical analysis comparing the subacute and chronic group does not appear. Therefore, perhaps the correct text as a comment on the table would be: ""The differences were significant in the chronic group (p = 0.021) when compared by cause of injury (traumatic or medical) but not when we compared the subacute group".

Response 4: We agree with this comment, we have changed the paragraph with your indications.

Comments 5: On lines 198-9, it says: "Statistically significant were the results when we compared groups (subacute and chronic) in State Anxiety (STAI-S)." WHAT STATISTICAL ANALYSIS DO YOU BASED ON TO STATE THIS? (I THINK IT DOES NOT APPEAR IN TABLE 2).

Response 5: This is an error; this comment is deleted.

Comments 6: In lines 224-5 it says: "About anxiety, it seems that the highest scores are obtained by the paraplegic group in both the acute and chronic stages." But the data do not support this statement in the acute phase, so the correct phrase should refer only to the chronic phase: "About anxiety, it seems that the highest scores are obtained by the paraplegic group in the chronic stage, although not significantly."

Response 6: We agree with this comment, therefore, we have put the full expression.

Comments 7: On line 233, it would be advisable to change the wording of the phrase: "Tables 6, 7 The results revealed a..." For example, there is a missing period followed by 7.

Response 7: We agree with this comment, therefore, we have changed this paragraph.

Comments 8: In Table 8, values ​​of p<0.05 could have an asterisk*, as in Table 1.

Response 8: We agree with this comment, therefore, we have put an asterisk.

Comments 9: IMPORTANT: On lines 239-240, it says: "The variables that are associated with suffering from state anxiety (STAI-S) in spinal cord injured people were that the cause of the injury was non-traumatic (OR=15.591... "   But it really seems that it correlates with TRAUMATICS. Therefore, I think the text should be: “The variables that are associated with suffering from state anxiety (STAI-S) in spinal cord injured people were that the cause of the injury was traumatic (OR=15.591..."

Response 9: We agree with this comment, this is an error, we have changed the paragraph.

Comments 10: In discussion, data appears that have not been previously clearly stated in Results. For example, on lines 272-3: "Our study shows results like those found, anxiety in the subacute group is at 8% STAI-T and 10% in STAI-S, and in the chronic group both STAI-T and STAI-S are at 14%".   

I propose that said data should previously appear in the Results section, and then be commented on in Discussion.

Response 10: We agree with this comment, we have included in the results these data.

Comments 11: IMPORTANT: Lines 279-282 should be DELETED (they are comments from the authors' draft): "Authors should discuss the results and how they can be interpreted from the perspective of previous studies and of the working hypotheses. The findings and their implications should be discussed in the broadest context possible. Future research directions may also be highlighted".

Response 11: We agree with this comment, we have deleted this paragraph.

Comments 12: IMPORTANT: In lines 334-5, in Conclusions, where it says: "The group of paraplegics and tetraplegics score higher than the group of subacute paraplegics, and these results are reversed in the chronic group".

It should say: "In the subacute group, the group of paraplegics score higher than the group of tetraplegics, and these results are reversed in the chronic group".

Response 12: We agree with this comment, therefore, we have put the full expression.

Comments 13: In lines 337-8, it says: "and this relationship is reversed in the chronic group". I think that final sentence should be removed, because the scores between both groups in the chronic cases are almost equal."

Response 13: We agree with this comment, therefore, we have removed this sentence.

Comments 14:  In conclusions 1, on Depression, I think it should specify which differences are significant and which are not.

Response 14: We agree with this comment, therefore, we have included these significant differences.

Comments 15: The last paragraph of the Conclusions is not derived from the results of this research and, therefore, I think it should appear in Discussion: "A correct mental state is necessary for the success of rehabilitation as well as for the ability to perform activities of daily living according to their disability. It is also essential for the planning and solving of problems in order to achieve a right social adaptation and, consequently, progress in their care, which will help to improve their quality of life."

Response 15: We agree with this comment, therefore, we have removed this paragraph in Conclusions and we have add it, in Discussion.